# Assessment of Real-Time Active Noise Control Devices in Dental Treatment Conditions

**DOI:** 10.3390/ijerph19159417

**Published:** 2022-08-01

**Authors:** Ik-Hwan Kim, Hyeonmin Cho, Je Seon Song, Wonse Park, Yooseok Shin, Ko Eun Lee

**Affiliations:** 1Department of Pediatric Dentistry, College of Dentistry, Yonsei University, Seoul 03722, Korea; kih86007@yuhs.ac (I.-H.K.); jhm7086@daum.net (H.C.); 2Oral Science Research Center, College of Dentistry, Yonsei University, Seoul 03722, Korea; densys@yuhs.ac; 3Department of Advanced General Dentistry, College of Dentistry, Yonsei University, Seoul 03722, Korea; wonse@yuhs.ac; 4Department of Conservative Dentistry, College of Dentistry, Yonsei University, Seoul 03722, Korea; 5Department of Pediatric Dentistry, Kyung Hee University Dental Hospital, Seoul 02447, Korea; olivedlr@naver.com

**Keywords:** dental noise, noise, noise control, noise hazard

## Abstract

Dental clinics are exposed to various uncomfortable noises. The aim of this study was to quantify the effectiveness of active noise control devices in dental treatment conditions. Two types of commercial headsets (Airpods Pro, QC30) and two types of dental headsets (Alltalk, Quieton Dental) were used for the experiment. Three sounds (high-speed handpiece, low-speed handpiece, and suction system) were measured at three different distances from the dental teeth model, typodont. The distances of 10, 40, and 70 cm reflected the positions of the patient, assistant, and practitioner’s ears, respectively. Sound analysis was performed, and the significance of differences in the maximum noise level using each device was determined with the Kruskal–Wallis test. Dental noise was characterized by the peak in sound pressure level (SPL) at 4–5 kHz and >15 kHz frequencies. The commercial headsets efficiently blocked 1 kHz and 10 kHz of noise. The dental headsets efficiently reduced 4–6 and >15 kHz noise. Quieton had the highest maximum SPL in all situations and positions among the four devices. For a better dental clinic, however, active noise control devices more suitable for the characteristics of dental noise should be developed.

## 1. Introduction

Noise is an unwanted sound that makes people uncomfortable. It is physically defined as a sound pressure level of 50 dB or more. Noise interferes with communication and is related to impaired personal abilities and concentration, behavioral, learning, and productivity limitations. In addition, noise causes a secondary stress response in humans and causes numerous diseases, such as sleep disorders, high blood pressure, greater risk of heart attack, and mental disorders [1].

Various noises are generated in the field of dentistry. Because pediatric dentists provide treatments, including restoration, orthodontics, surgical operation, etc., there is a lot of dental equipment making high-frequency noises. For example, noise from high and low-speed handpiece drills makes 70–92 dB of noise [2,3,4,5] and ultrasonic scalers make 68–88 dB of noise [2,3,6]. Screaming and crying from younger patients reaches about a 110 dB level [2,7]. Due to the limited treatment time, no hearing loss occurs. However, there are reports that dental noise is associated with fear and causes discomfort to patients and dental staff. Noise also interferes with the communication between medical staff and between patients and staff.

Therefore, properly blocking dental noise can alleviate patients’ fear and improve treatment quality. However, excessive noise canceling can also interfere with the general dialogue between patients–dentists–guardians and decrease satisfaction due to difficulties in communication during treatment. Thus, a meticulous strategy to efficiently block dental noise is required. Earplugs and headset-type hearing protection equipment used in construction sites and indoor work processes easily block noise, but they are inappropriate for dentistry because they interfere with daily communication.

Active noise control (ANC) is literally cancelling or amplifying sound as intended by the developer. The ultimate goal of ANC is to reduce unwanted noise and maintain or amplify only the targeted sound. These techniques were first presented in the 1930s, and attempts to apply them to headsets began in the 1950s [8]. ANC was initially developed for aircraft passengers and attempted to reduce noise in automobiles and is now being applied to simple music listening. ANC uses reference microphones to selectively block or amplify sounds. The reference microphones provide the waveform of unwanted noise and sounds that they want to amplify.

Noise has a cancellation effect by emitting an inverse phase waveform of noise and amplifying the desired sound. Passive noise control (PNC), unlike ANC, is the effect of blocking waves through mechanical physical shields, including sound absorbers and the foam in sound devices. In addition, active filtering (AF) refers to the effect of changing the waveform at the electrical signal stage. ANC, PNC, and AF come up with a strategy to block noise by combining the three, which is collectively referred to as ANC [9]. We have identified the characteristics of the equipment used in this study based on the fact that PNC easily blocks high-frequency noise at 10 kHz or higher and ANC easily blocks noise that is relatively lower than 10 kHz.

ANC has been introduced by several sound equipment companies and applied on commercial headsets, which use external reference microphones to obtain noise waveforms and generate opposite waveforms to offset noise [9]. The purpose is not just to physically block noise, but to create an environment where you can listen to the desired sound well. At the same time, the desired sound is also emitted into the headset so that it can be heard, thus achieving two objectives: communication and noise blocking. Existing commercial products focus on blocking public transportation sounds or other noises other than music output, but noise in daily life and dental clinics have different characteristics, so the effectiveness of using commercial products in dental clinics is not guaranteed.

In this study, we made a hypothesis that the dental products are more advantageous for dental noise than commercial ones. By reproducing a dental noise environment, dental products and commercial products were compared.

Recently, dental-noise-specific ANC products are starting to appear on the market, but comparative studies of commercial and dental products are absent. In response, this study aims to compare the usefulness of commercial and dental products under a strictly designed experimental environment and to emphasize the need to develop better dental ANC equipment in the future.

## 2. Materials and Methods

### 2.1. Devices

The 3-DIO free space (3DIO, Vancouver, BC, Canada) is a binaural microphone, and Tascam DR44WL (Tascam, Santa Fe springs, CA, USA) is a record controller (Figure 1). High-speed and low-speed handpiece models are GENTLEmini LUX 5000B (Kavo, Algan, Germany) and INTRAmatic 20CN (Kavo, Algan, Germany). Additionally, the suction system from an A-dec 300 (A-dec, Newberg, OR, USA) dental chair was used for the experiment.

Four noise-cancelling devices were used. Airpods Pro (Apple, Cupertino, CA, USA) and QC30 (Bose, Framingham, MA, USA) are commercial headsets with an active noise control property. Alltalk (Healing sound, Seoul, Korea) and Quieton dental (Quieton, Kempele, Finland) are dental-specific active noise control devices.

### 2.2. Specification of Experimental Place

This study was performed in an isolated dental office. The width and length of the dental office were 3.82 m and 4.51 m each. There were no structures such as soundproofing materials on the ceiling and walls in the dental office, and it was a general concrete structure. Height was 2.86 m with a 1.46 m tall partition on the lower side and with a 1.4 m tall sealed sash on the upper side. Dental unit chair and other equipment were not moved, and the environment was the same as on normal workdays.

### 2.3. Recording Procedure

The procedure was conducted at 7:00 PM after the end of the treatment hours. The dental unit chair was set in supine position, and typodont with artificial teeth was set on the headrest. Position of microphone was divided into three. Each position reflects environments of practitioner, dental assistant, and patient. Distance between teeth and microphone in each position is 40 (for practitioner), 70 (for dental assistant), and 10 (for patient) cm. At each position, cases vary, as shown in Table 1. Both #330 high-speed bur and #6 low-speed round bur were prepared for tooth preparation. Artificial resin teeth (lower right second primary molar) with class I cavity on occlusal surface were also prepared. In preparation cases, the cavity wall was reduced with a handpiece under water spraying during recording (in suctioning situation, suctioning of sprayed water is regarded the same as preparation). In each case, recording lasted for 10 s, and the procedure was repeated five times.

Each recording was repeated five times in the same manner. The conditions listed above were repeatedly implemented according to the position of the microphone.

### 2.4. Sound Analysis

To acquire maximum sound level and frequency of files, an algorithm was coded with Fortran language on AcroEdit (Acrosoft, Seoul, Korea) program. An executable file was made with Intel Fortran Complier (Intel, Santa Clara, CA, USA) using the code. One file that included less unwanted noise was selected among five repeated files. To obtain a frequency (Hz)–sound pressure level (SPL, dB) graph of each file, Tecplot 360 (Tecplot, Bellevue, WA, USA) was used. The moment 5 s after the start of recording was captured and visualized with Hz–dB graph. At that moment, the highest point of the graph represents maximum SPL.

### 2.5. Comparison of Graphs

As previously presented in Table 1, numbered conditions were compared. Qualitative comparison was based on their morphology of graphs and quantitative comparison was based on statistical analysis of maximum SPL.

### 2.6. Statistical Analysis

Statistical comparison of maximum SPLs (dB) of each file were carried out using SPSS (version 23.0.0, SPSS, Chicago, IL, USA) and Excel 2010 (Microsoft Inc., Chicago, IL, USA). Kruskal–Wallis test was performed to determine the significance of differences in maximum SPL using each device. A Mann–Whitney test was performed for post-hoc using level of significance (*p* < 0.05) by Bonferroni’s method.

## 3. Results

A total of 525 files in MP3 format were obtained by repeatedly measuring 35 cases at three locations five times.

### 3.1. Characteristics of Dental Noise

Compared to the absence of tooth preparation (Figure 2), the noise increased in the tooth preparation situation (Figure 3), with the average increase being the highest for the operator and the lowest for suctioning. In particular, when a high-speed drill was used for tooth preparation, the noise increased at the operator’s position, but decreased on average at the assistant and patient positions. Qualitatively assessed on the graph, when idling without tooth preparation the noise was only high at certain frequencies, but with tooth preparation SPLs were found to rise at all frequencies.

With the high-speed handpiece, the SPL was the highest at frequencies of 6 kHz and 19 kHz in non-preparation cases. When the handpiece works, noise was reduced at 19 kHz and increased at other frequencies. The SPL was the highest in the patient position and the lowest in the assistant position. With the low-speed handpiece, the SPL was the highest at 4–5 kHz, but the overall SPL was more equivalent at all of the frequencies other than high-speed. The SPL was the highest in the patient position and the lowest in the assistant position. The difference caused by the tooth preparation was not noticeable. During suctioning, the SPL was high at all frequencies in plateau form (5–10 kHz). Uniquely, noise decreased during water spraying and aspiration at higher frequency.

### 3.2. Comparison of Each Device Character

In the case of the Airpods Pro, the sound cancellation capability was generally excellent, but an excessive sound cancellation effect was also shown at 1 kHz, which may hinder normal conversations or voices. Additionally, the noise of 5 kHz at high-speed was not effectively suppressed. Since it is a commercial product for listening to music, music is heard smoothly (4 kHz) and it is programmed to block urban noise such as human voices (1 kHz) and noise from public transportation (10 kHz). The cancellation of 17 kHz noise is believed to be blocked by the foam in the Airpods Pro. Alltalk showed no cancellation in the 1 kHz region and optional cancellation in the 4 kHz region. However, near 5 kHz, which is the main noise region of dental instruments such as high-speed handpieces, the minimum canceling effect was shown which was not effective in reducing actual noise. However, the noise cancelling effect on high-frequency noise that causes irritation was excellent. QC30 seems to attempt more selective sound cancellation compared to Airpods Pro. Noise of 0–3 and 8–10 kHz was blocked, like in the Airpods Pro, but others were mostly maintained. The disadvantage of QC30 is that the sound cancellation is quite weak in the high-frequency area of 15 kHz and over. Quieton dental is a device that can effectively block dental noise by blocking noise above 3 kHz without disrupting normal conversations by maintaining or amplifying sound under 3 kHz. However, the results of this study did not show any relation or trend between maximum SPL reduction.

Focusing on the characteristics of each piece of equipment (Figure 4 and Figure 5), the Airpods Pro mainly cancelled noise around 10, 17 kHz, maintained or amplified noise at 4 kHz, and slightly decreased noise at 0–3 kHz and 13 kHz. Alltalk mainly cancelled noise around 10, 15 kHz, amplified at 1, 5 kHz, with a slight drop at 4, 12, 17 kHz. QC30 intensively cancelled noise at 8–10 kHz and maintained or amplified at 5,12 kHz. Instead of having uniform sound cancellation overall, Quieton maintained or amplified the noise at 0–3 kHz and showed the best sound cancellation effect in the higher frequency domain.

### 3.3. Comparison of Various Situations

Noise cancellation was not significant at peaks of certain frequencies that occur in the non-preparation situation (Figure 4). In the preparation situation (Figure 5), the SPL was relatively similar at all frequencies, and the noise cancellation effect was relatively equal compared to the non-preparation situation. There was a difference in the size of the noise according to the recording distance. Although the characteristics of noise cancelling for each device do not change, the closer the distance is, the less the effect is. On the graphs, the characteristics of each device did not change depending on the situation.

### 3.4. Comparison of Maximum Noise Level

Each recorded file’s maximum SPL and corresponding frequency were compared. With Quieton, it has the lowest frequency, regardless of the situation and distance, and it always has a higher noise level than the control. When the high-speed handpiece is operated, all devices make the frequency much lower than the control. Only in the situation of suctioning could the frequency be reduced by any devices. When using the low-speed handpiece, all devices except Alltalk had the effect of decreasing the frequency. There was no consistency or trendability in the maximum SPL change statistically. Each maximum SPL is sorted in Table 2.

## 4. Discussion

Noise is one of the indicators of a pleasant environment and is emerging as a concern in daily life. In particular, continuous exposure to noise can harm health, such as leading ischemic heart disease, stroke, and high blood pressure. Noise generated from equipment in dental clinics can interfere with treatment and communication and is one of the causes of fear in dental treatment. In addition, dental workers are always exposed to a noisy environment, so it is easy to increase stress and decrease concentration due to noise.

All four devices blocked low-speed handpiece and suction noise much better than high-speed handpiece noise. However, the maximum SPL of low-speed handpiece noise was higher than that for high-speed using all of the devices. Focusing on the peaks in the frequency–SPL graphs, the peak of the low-speed handpiece is under 5 kHz frequency. On the other hand, the peak of the high-speed handpiece is 5 kHz and over. The reason why the high-speed handpiece noise is more easily blocked is supported by the fact that noise below 5 kHz was more difficult to block than the noise above 5 kHz for all of the devices in this study. The peak with the maximum SPL of the low-speed handpiece is 4–5 kHz, which is not properly blocked, or rather, is amplified. The high-speed handpiece shows a decrease in peak levels, including the effect of lowering the frequency, such as the peak at 6 kHz, which makes the overall noise reduction effect for the low-speed handpiece greater, but the main noise can be easily cut off for the high-speed handpiece. Normal conversation sounds (near 1 kHz) were better maintained mainly with dental devices (Alltalk, Quieton), which were more effective in cutting off high-frequency noise.

As a result of analyzing the noise that can be generated during dental treatment, the effort to spatially isolate a procedure that can generate a specific noise can help reduce noise in the dental clinic. Additionally, it is necessary to develop a noise-canceling device to improve communication between patients and staff and to reduce patient discomfort, but efforts are also needed to protect the work environment of staff continuously exposed to noise.

Selective sound control is dependent on the purpose, and individual strategies must be established to eliminate the certain type of noise. Daily life noise such as public transportation (10 kHz) or human speech (1 kHz) should be blocked for simple listening to music, but sounds near 1 kHz, where hearing loss occurs first, should be actively blocked to prevent aging-induced hearing loss. For preventing noise-induced hearing loss caused by dental noise, the 4–6 kHz region and high-frequency noise above 10 kHz should be blocked rather than daily life noise. At the same time, 1 kHz of sound should be maintained or amplified because verbal conversations must be possible in the dental clinic. As the distance becomes smaller, the device’s ability decreases, so it is expected that more high-performance sound control device will be required for equipment used for patients who feel the noise closest to them.

The hypothesis that dental noise affects staff’s auditory acuity is controversial. The hearing threshold of dentists did not rise to a statistical difference compared to that of non-dentists, dismissing it as only temporary [6,10,11,12]. In particular, it is said to be insignificant compared to the conditions outside of the clinic, including in the laboratory and the sterilization room. On the other hand, there is a tendency for hearing impairment to affect dental workers, particularly in the high-frequency range of 4, 6, and 8 kHz [13,14,15]. As a result, dental staff’s audio frequencies tended to become narrower the longer their careers, especially in men [16], supporting the opinion that the dental environment has a ontological effect on hearing loss. Moreover, dental noise is a high-intensity noise that can cause temporary hearing changes, and patients’ screams are believed to cause acoustic trauma [17]. In addition to the effects on hearing, noise also: increases blood pressure, heart rate, and urination; reduces adrenal cortex hormones; and hinders communication in the clinic, making it difficult for dentists and patients to communicate with each other. Therefore, efforts to control dental noise are necessary.

Noise generated in the workplace increases the stress of workers and acts as a major cause of psychological anxiety and hearing impairment. In general, attention is paid to periodic medical examination and the management of noise-induced hearing loss in noise-exposed workers, but less attention is paid to tinnitus. In particular, tinnitus is often accompanied by hearing loss, and it can cause physical disability along with hearing loss. Additionally, caution is required because tinnitus itself has a profound effect on activities and quality of life in daily life, and it is difficult to identify and treat the cause. Noise control in dental clinics is necessary to study long-term to protect the hearing of workers due to continuous noise exposure.

Human beings also lose their hearing ability through aging. Aging-induced hearing loss is characterized by a decrease in the high-frequency area (15–20 kHz) within the hearing frequency range and a decrease in hearing at the frequency (1 kHz) of the conversation area. Noise-induced hearing loss, on the other hand, starts at around 4 kHz and gradually expands to 1–8 kHz. Therefore, the fact that commercial products are not set to cancel 4 kHz sound for listening to music is disappointing from the view of noise-induced hearing loss prevention. If dental devices are advantageous in preventing noise-induced hearing loss by showing sound cancelling ability in the 4–5 kHz area, commercial devices can be judged to further prevent aging-induced hearing loss through 1 kHz sound cancellation. Since all devices have shown sound cancellation ability in high-frequency areas, it is also meaningful in protecting the human audio frequency area. In particular, it is judged to be utilizable in communication, such as improved speech recognition in ANC environment than in PNC alone [18] and frequency of readback when using ANC headphones in aircraft also decreased [19,20]. However, in order to prevent both noise- and aging-induced hearing loss while improving communication at the same time, the SPL should be able to be controlled by frequency more precisely than the current method.

There is a limit to qualitative evaluation alone. Quantitative analysis of frequencies and SPL is omitted, as this study is an experiment in which the results are based on the data subjectively determined through the form of graphs. The evaluation of each device felt by an individual may be significantly different from that of the author. Nevertheless, this study is meaningful because it first compared the ANC property of commercial equipment with dental equipment in a similar dental noise environment. Comparison of several variables will be required in subsequent studies through precise analysis in a more controlled experimental environment. As other articles have suggested, the degree of fitting of experimental devices (especially foam) to the binaural microphone depends on noise cancellation performance [21], so better research results can be obtained if the uniformity in the shape of the artificial ear model and device was also reflected.

## 5. Conclusions

Airpods Pro and QC30 had strengths in blocking noise at 1, 10 kHz, which means they block human voices and public transportation noise, suggesting that there is a limit to blocking dental noise at 4–6 kHz. On the other hand, the Alltalk and Quieton dental devices had the advantage of cancelling noise at 5–6 kHz and more than 15 kHz, while maintaining or amplifying noise at 0–4 kHz, so they do not interfere with human conversation. Dental products were more useful in the dental office than other commercial products according to these results. However, it is difficult to compare performance of devices clearly because the statistical trend was ambiguous. Devices can also have the effect of making noise feel rather loud because it maintains or amplifies noise below 5 kHz, which is easy for humans to hear. ANC equipment that reflects the characteristics of dental noise more precisely but does not interfere with conversation should be developed. More precisely designed dental products will improve the quality of dental treatment and provide comfort to people in the clinic.

## Figures and Tables

**Figure 1 ijerph-19-09417-f001:**
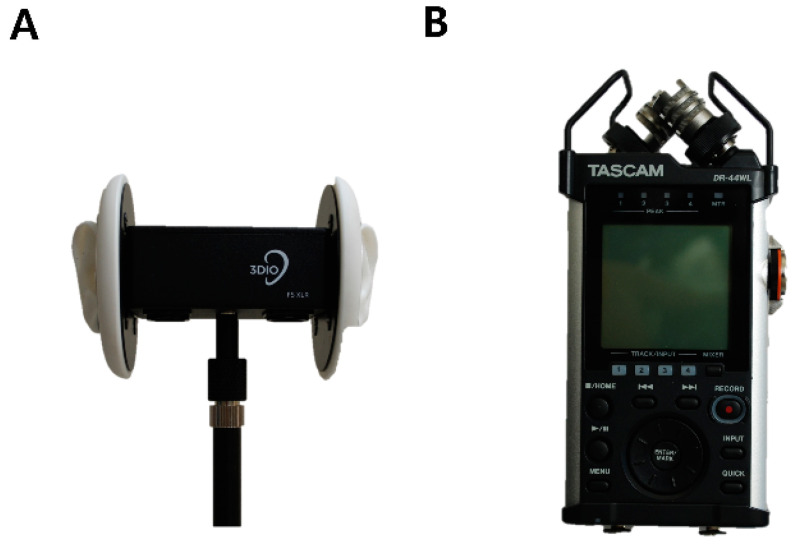
Equipment used for recording: (**A**) Binaural microphone Free space (3DIO, Vancouver, Canada); (**B**) Record controller DR44WL (Tascam, Santa Fe springs, CA, USA).

**Figure 2 ijerph-19-09417-f002:**
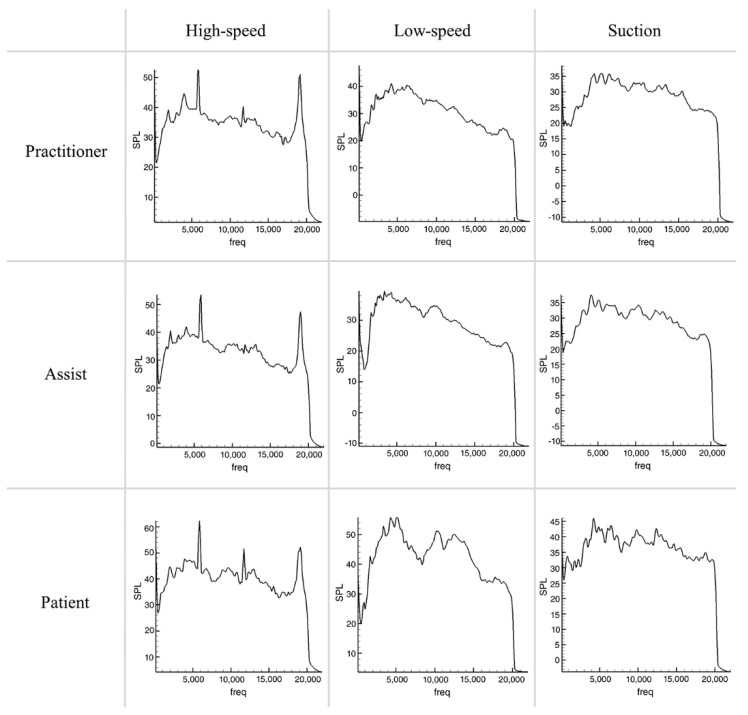
Sound pressure level graphs of dental noise from high−speed, low−speed, and suction without tooth preparation.

**Figure 3 ijerph-19-09417-f003:**
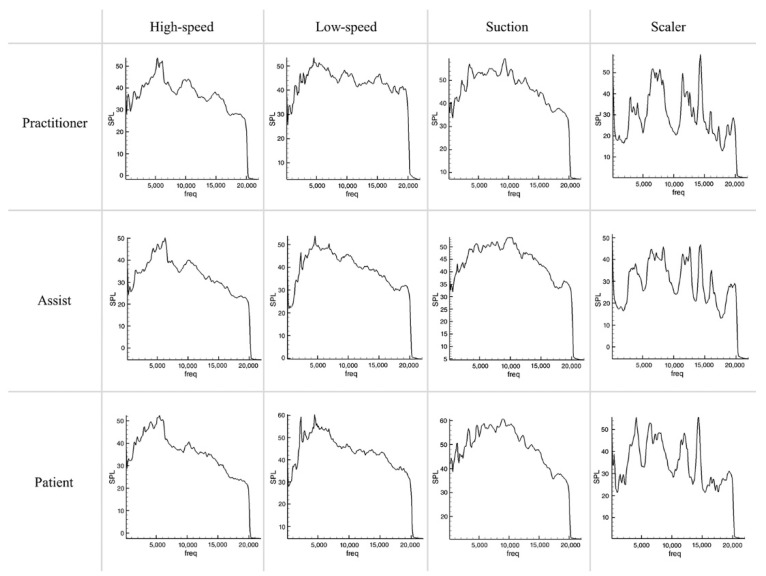
Sound pressure level graphs of dental noise from high−speed, low−speed, suction, and scaler during tooth preparation.

**Figure 4 ijerph-19-09417-f004:**
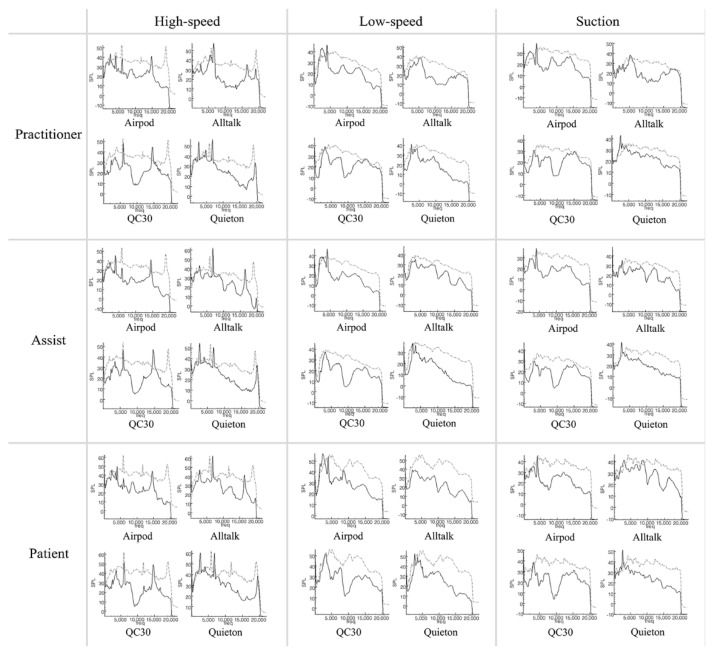
Comparison of sound pressure level not during tooth preparation. Dash−dot graph represents control condition (dental noise) and solid graph represents experimental condition (active noise control devices).

**Figure 5 ijerph-19-09417-f005:**
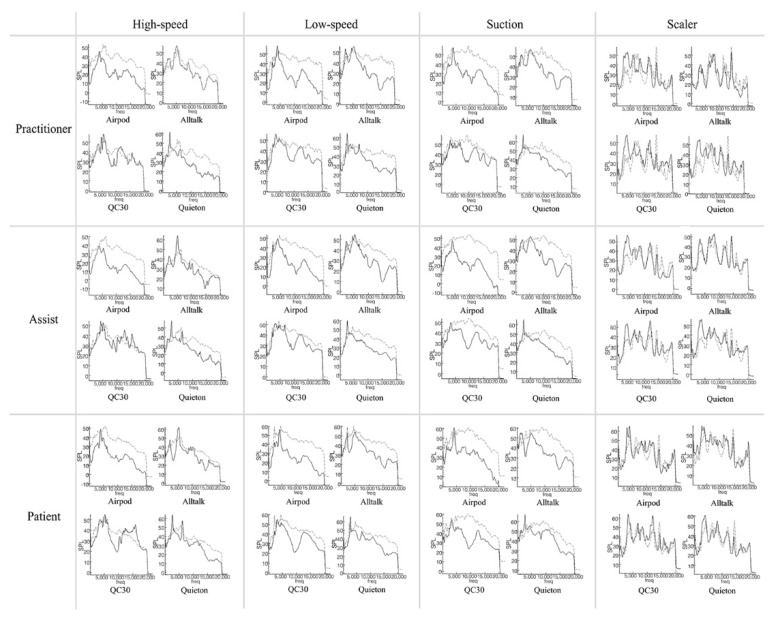
Comparison of sound pressure level during tooth preparation. Dash−dot graph represents control condition (dental noise) and solid graph represents experimental condition (active noise control devices).

**Table 1 ijerph-19-09417-t001:** Conditions of the recordings.

		Control	Air Pod Pro	Alltalk	QC30	Quieton
Without Tooth preparation	High-speed	No. 1	No. 2	No. 3	No. 4	No. 5
	Low-speed	No. 6	No. 7	No. 8	No. 9	No. 10
	Suction	No. 11	No. 12	No. 13	No. 14	No. 15
With Tooth preparation	High-speed	No. 16	No. 17	No. 18	No. 19	No. 20
	Low-speed	No. 21	No. 22	No. 23	No. 24	No. 25
	Suction	No. 26	No. 27	No. 28	No. 29	No. 30
	Scaler	No. 31	No. 32	No. 33	No. 34	No. 35

**Table 2 ijerph-19-09417-t002:** Mean value of maximum sound pressure level (during tooth preparation).

	**High**					
	**Control**	**Airpods Pro**	**Alltalk**	**QC30**	**Quieton**	** *p* **
Practitioner	53.55 ^ab^	48.41 ^a^	57.37 ^abc^	58.26 ^bc^	61.41 ^c^	0.000
Assistant	50.45 ^ab^	39.18 ^a^	64.27 ^c^	54.43 ^abc^	58.37 ^bc^	0.000
Patient	52.57 ^ab^	48.50 ^a^	62.38 ^bc^	54.61 ^abc^	64.37 ^c^	0.000
	**Low**					
	**Control**	**Airpods Pro**	**Alltalk**	**QC30**	**Quieton**	** *p* **
Practitioner	53.85 ^a^	58.45 ^bc^	55.54 ^ab^	56.21 ^abc^	64.04 ^c^	0.000
Assistant	53.55 ^abc^	49.62 ^a^	54.68 ^bc^	51.22 ^ab^	60.11 ^c^	0.000
Patient	60.39 ^bc^	56.2 ^abc^	54.38 ^a^	55.19 ^ab^	66.12 ^c^	0.000
	**Suction**					
	**Control**	**Airpods Pro**	**Alltalk**	**QC30**	**Quieton**	** *p* **
Practitioner	59.67 ^bc^	57.48 ^abc^	57.15 ^ab^	54.6 ^a^	68.38 ^c^	0.000
Assistant	53.76 ^bc^	47.52 ^a^	53.23 ^abc^	49.82 ^ab^	65.44 ^c^	0.000
Patient	60.71 ^ab^	60.59 ^ab^	55.19 ^a^	53.76 ^a^	67.3 ^b^	0.000

*p* value from Kruskal–Wallis test. ^a,b,c^: The same character means no statistical difference by Mann–Whitney test.

## Data Availability

Not applicable.

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
