# Peer review of "Assessment of Real-Time Active Noise Control Devices in Dental Treatment Conditions"

_ijerph, 2022, doi:10.3390/ijerph19159417_

Round 1

Reviewer 1 Report

This is an interesting topic, well-designed study and well-written paper.

The authors could make it more clear as to the reason for studying noise reduction: is it (a) to reduce patient anxiety from the noise in their mouth during treatment or (b) to reduce patient anxiety from hearing the noise in the clinic or (c) improve the conditions in the workplace for staff?  This distinction is important as to the purpose of the paper which is unclear.

The study design is good however in the real situation there is bone conduction of sound which greatly reduces the benefit of ANC devices.  We know that QuietOn Dental is ineffective in the clinic for patients but might help staff.

In the discussion the authors should make it clear than there may be a benefit for clinical staff but there is no implied benefit to patients from drill noise, which is a major cause of anxiety.

Line 191 needs a correction to the English.

There are a few relevant references from the E Kaymak et al group that are missing.  Reference 8 is incomplete. 

Overall I congratulate the authors on a good manuscript and hope to see it published with these minor adjustments. 

Author Response

First of all, thank you for the nice comments and guidance.

The authors could make it more clear as to the reason for studying noise reduction: is it (a) to reduce patient anxiety from the noise in their mouth during treatment or (b) to reduce patient anxiety from hearing the noise in the clinic or (c) improve the conditions in the workplace for staff?  This distinction is important as to the purpose of the paper which is unclear.

  • Although this study aims at all of the points (a), (b), and (c) you pointed out, in this study, (a) is considered to be the most important point. This part has been corrected and the introduction has been supplemented.

The study design is good however in the real situation there is bone conduction of sound which greatly reduces the benefit of ANC devices.  We know that QuietOn Dental is ineffective in the clinic for patients but might help staff.

  • Thanks for the great comment.

In the discussion the authors should make it clear than there may be a benefit for clinical staff but there is no implied benefit to patients from drill noise, which is a major cause of anxiety.

  • If the ANC device is used, it is judged to be effective for the clinical staff and patients as well as the drill sound is reduced. It is necessary to develop and research a device that does not reduce the sound of everyday conversation, but only reduces the specific noise of dental instruments, which is expected to improve communication between patients and doctors and reduce patients' fear of dental treatment.

Line 191 needs a correction to the English.

  • Thank you for your detailed insight. Correction has been completed.

There are a few relevant references from the E Kaymak et al group that are missing.  Reference 8 is incomplete. 

Thank you for your detailed insight. Correction has been completed.

Reviewer 2 Report

Dear Authors first of all I would like to congratulate you for your work, It is a good research. I would like to highlight some points in addition to this:

I would prefer that you would made a longer introduction, with more data on the noise type that occurs in the dental clinic. I suggest that you make an introduction longer.

In the discussion you present results and background on the active noise reduction, the first part of the discussion should be on results and all that you explain on the active noise reduction should be on the introduction.

Author Response

First of all, thank you for the nice comments and guidance.

I would prefer that you would made a longer introduction, with more data on the noise type that occurs in the dental clinic. I suggest that you make an introduction longer. In the discussion you present results and background on the active noise reduction, the first part of the discussion should be on results and all that you explain on the active noise reduction should be on the introduction.

  • Thank you for your detailed insight. Correction has been completed.

Reviewer 3 Report

The authors aimed to compare the usefulness of commercial and dental products under  strictly designed experimental environment and to emphasize the need to develop better  dental Active noise control (ANC) equipment in the future.

The study covers some issues that have been overlooked in other similar topics. The structure of the manuscript appears adequate and well divided in the sections. Moreover, the study is easy to follow, but some issues should be improved. Some of the comments that would improve the overall quality of the study are:

a. Authors must pay attention to the technical terms acronyms they used in the text.

b. English language needs to be revised.                              

c. Conclusion Section: This paragraph required a general revision to eliminate redundant sentences and to add some "take-home message".

Author Response

First of all, thank you for the nice comments and guidance.

  1. Authors must pay attention to the technical terms acronyms they used in the text.
  2. English language needs to be revised.                              
  3. Conclusion Section: This paragraph required a general revision to eliminate redundant sentences and to add some "take-home message".
  • Thank you for your detailed insight. Correction has been completed.

Reviewer 4 Report

Line 16    Please add some information about the original reason for this research, answer to question: why?

Line 17     Why this choice of headsets, no more available, or something else?

Line 19    Typodont, please add: dental teeth model.

Line 25    The commercial headsets …… English is not correct and sentence is not complete!

Line 34    Definition of sound is a little too “thin”, it is a combination of the two explanations: “uncomfortable” and too loud (above 50 dB).

Line 35    “In medically ….” (sentence) English is not correct, please adjust!

Line 41    140 dB is extreme, screaming reaches on not more than 100-110 dB, if it is nearby at most 120 dB

Line 46    It seems to me that “careless noise cancelling” is meant in this sentence ….. is that true?

Line 81    Figure 1A is all right, but figure 1B seems a little overdone, the figure does not add new information.

Line 85    Please add some information about acoustics of dental office, walls/ceiling with/without absorption materials?

Line 89    7 PM, why this time, why is it necessary to mention?

Line 100  Table 1: it is not clear to me why it should be necessary to mention the exact number of the recordings?

Line 126   It is not clear what is meant by 0.97-24.04 dB, please adjust or explain!

Line 129   It is not clear what is meant by in an increase of only 0.97 dB?

Line 135   “….other not during…..” is not correct English, please adjust!

Line 182   Table 2: maximum sound pressure level: how to measure, averaged over (short) time?

Line 186  ….which has the disadvantage of …. Is not correct English, one could image the opposite!

Line 200  “trendability” is not the right word, it should be explained in another way, English weak!

Line 212   “is greater” in this sentence, is not correct English, please adjust (e.g. omit “is”).

Line 278  278 meaningfull should be preceded by “is”.

Tinnitus is not mentioned as a hearing problem, did it occur?Presumably it should be reported in the same situation?

Author Response

First of all, thank you for the nice comments and guidance.

Line 16    Please add some information about the original reason for this research, answer to question: why?

Line 17     Why this choice of headsets, no more available, or something else?

  • We selected headsets that allow conversation while wearing it.

Line 19    Typodont, please add: dental teeth model.

Line 25    The commercial headsets …… English is not correct and sentence is not complete!

  • Thank you for your precious advice. The shortcomings of the abstract were corrected and a full complement was implemented.

Line 34    Definition of sound is a little too “thin”, it is a combination of the two explanations: “uncomfortable” and too loud (above 50 dB).

Line 35    “In medically ….” (sentence) English is not correct, please adjust!

Line 41    140 dB is extreme, screaming reaches on not more than 100-110 dB, if it is nearby at most 120 dB

Line 46    It seems to me that “careless noise cancelling” is meant in this sentence ….. is that true?

  • Thank you for your precious advice. I read the relevant part carefully, corrected the incorrect information, and organized the necessary information.

Line 81    Figure 1A is all right, but figure 1B seems a little overdone, the figure does not add new information.

  • This figure was created to provide information about the shape and model of the product. It is attached as an instrument that is not mainly used in dentistry. I will delete it if unnecessary.

Line 85    Please add some information about acoustics of dental office, walls/ceiling with/without absorption materials?

  • Thank you for your precious advice. Content has been added and modified.

Line 89    7 PM, why this time, why is it necessary to mention?

  • It means the time when the treatment is over, and it is written to mean that there is no noise from other unit chairs or patients.

Line 100  Table 1: it is not clear to me why it should be necessary to mention the exact number of the recordings?

  • Numbering was done as a means to distinguish each device and experiment. Separated for management of recorded files and results.

Line 126   It is not clear what is meant by 0.97-24.04 dB, please adjust or explain!

Line 129   It is not clear what is meant by in an increase of only 0.97 dB?

  • I tried to attach the sound analysis result data, but it was deleted because it was inappropriate for this study and figure.

Line 135   “….other not during…..” is not correct English, please adjust!

  • Thank you for your precious advice. Content has been added and modified.

Line 182   Table 2: maximum sound pressure level: how to measure, averaged over (short) time?

  • To get frequency (Hz)-Sound pressure level (SPL, dB) graph of each file, Tecplot 360 was used. The moment 5 seconds after the start of recording was captured and visualized with Hz-dB graph. At that moment, the highest point of graph represents maximum SPL.

Line 186  ….which has the disadvantage of …. Is not correct English, one could image the opposite!

Line 200  “trendability” is not the right word, it should be explained in another way, English weak!

  • Thank you for your precious advice. Content has been added and modified.
  • The content has been moved to the result part as pointed out by other reviewers.

Line 212   “is greater” in this sentence, is not correct English, please adjust (e.g. omit “is”).

Line 278  278 meaningfull should be preceded by “is”.

  • Thank you for your precious advice. Content has been added and modified.

Tinnitus is not mentioned as a hearing problem, did it occur?Presumably it should be reported in the same situation?

  • Thank you for your valuable comments. Tinnitus is also one of the side effects of noise. However, due to the limited treatment time, it did not occur in dentistry.

Round 2

Reviewer 2 Report

Dear authors. Now the article is ready to publish. Thank you

Author Response

Thank you for your review.

Reviewer 4 Report

See attachment!

Author Response

Line 17

It is still unclear why these headsets, that allow conversation while wearing it, has been chosen.

It should be more elegant to chose headsets that are proven to be “the best”.

  • In the case of commercial headset, airpod pro, which is the most sold product globally, and Bose, which is known for its excellent ANC function, were used. We thought it was the best product in terms of popularity and ANC capabilities for commercial headsets, so we used it for research.

Line 19

Typodont, please add in text: dental teeth model (because not everyone knows this).

  • Added to the revised manuscript.

Line 120

Now it is clear what is meant with the exact number of the recordings.

I understand that the subjects had to listen to 35 recordings.

Please, add some words about possible tiredness of subjects.

What is the order of presentation of the recordings?

To my opinion it should be a random order, to avoid effects of tiredness.

Please, add some words and explanation about this issue.

  • Thank you for your detailed comments and review. All recordings have different preparation procedures for the corresponding process, so after one preparation, five consecutive recordings were performed to minimize the influence of the surrounding environment. The recorded file is analyzed through the program, and it is judged that there is no significant correlation with the tiredness of the subject. The recording was conducted at a dental clinic, and the recorded files were analyzed elsewhere, so we believe that the continuous recording did not affect the results. Thank you for your careful review.

 Tinnitus is not mentioned. But tinnitus is one of the (important) side effects of noise, as you stated. That’s why it should be mentioned in the discussion to make clear that tinnitus is an important side effect but was not taken into account in this study due to …… (explanation)

  • This study is an analysis study on the device related to the effective blocking of noise generated in the dental treatment environment. Tinnitus is an important side effect that can be caused by noise, but it is not a situation that can occur to the experimenter in this study, and it was not confirmed as it is not a subject to judge whether the experimenter has tinnitus and auditory abnormalities. However, as you said, tinnitus is highly related to noise, so we have added related content to the discussion. Thank you.
